

# Learning about the vertical structure of radar reflectivity using hydrometeor classes and neural networks in the Swiss Alps

Floor van den Heuvel[1,2], Loris Foresti[2], Marco Gabella[2], Urs Germann[2], and Alexis Berne[1]

[1]Ecole Polytechnique Fédérale de Lausanne (EPFL), Lausanne, Switzerland
[2]Swiss Federal Office of Meteorology and Climatology (MeteoSwiss), Locarno-Monti, Switzerland

**Correspondence:** Alexis Berne (alexis.berne@epfl.ch)

**Abstract.**

The use of radar for precipitation measurement in mountainous regions is complicated by many factors, especially beam shielding by terrain features, which, for example, reduces the visibility of the shallow precipitation systems during the cold season. When extrapolating the radar measurements aloft for quantitative precipitation estimation (QPE) at the ground, these

must be corrected for the vertical change of the radar echo caused by the growth and transformation of precipitation. Building on the availability of polarimetric data and a hydrometeor classification algorithm, this work explores the potential of machine learning methods to study the vertical structure of precipitation in Switzerland and to propose a more localised vertical profile correction. It first establishes the ground work for the use of machine learning methods in this context: from volumetric data of 30 precipitation events vertical cones with 500 m vertical resolution are extracted. It is shown that these cones can well represent

the vertical structure of different types of precipitation events (stratiform, convective, snowfall). The reflectivity data and the hydrometeor proportions from the extracted cones constitute the input for the training of artificial neural networks (ANN), which are used to predict the vertical change in reflectivity. Lower height levels are gradually removed in order to test the ANN's ability to extrapolate the radar measurements to the ground level. It is found that ANN models using the information on hydrometeor proportions can predict from altitudes between 500 and 1000 metres higher than the ANN based on only

reflectivity data. In comparison with more traditional vertical profile correction techniques the ANNs show less prediction errors made from all height levels up to 4000 m a.s.l., above which the ANNs lose predictive skill and the performance levels off to a constant value.

## 1 Introduction

Precipitation constitutes a key meteorological variable for ecosystems and societies; both as a primary input for freshwater

resources and (in deficit or excess) as a potential threat to infrastructure and human lives. Mountainous regions such as the Alps, through their impact on the flow and stability of air masses, influence the spatial distribution of precipitation (Frei and Schär, 1998; Roe, 2005; Colle et al., 2013) as well as precipitation growth processes and microphysics (Yuter and Houze, 2003; Colle et al., 2005; Stoelinga et al., 2013).





Nevertheless, measuring precipitation in the Alps remains a challenge. Ground raingauge networks are typically affected by poorer spatial representativity, wind-induced errors (especially in the case of solid precipitation, Nitu et al., 2018) and practical difficulties related to access and maintenance. On the other hand, quantitative precipitation estimates (QPE) by radar are also limited by many factors (e.g. Germann and Joss, 2002; Villarini and Krajewski, 2010). While the errors induced by ground

clutter, hardware calibration and - to a certain extent - attenuation (Germann, 2000; Germann et al., 2006, 2015; Gabella et al., 2016), can be satisfactorily dealt with within the Swiss polarimetric C-band radar network, reduced visibility and the correction for the changes in the vertical profile of precipitation remain important challenges (Jordan et al., 2000; Germann and Joss, 2002). Visibility reduction due to partial and total beam shielding by mountainous terrain is partly overcome by the higher elevation locations of the Swiss radars (between ∼900 and ∼3000 m a.s.l.) but this also exacerbates the effects of overshooting

due to Earth curvature. As a result, the radars cannot measure in the lowest layers of the atmosphere. For areas with reduced visibility it is common practice to use radar measurements from aloft to estimate precipitation quantities at the ground level. These measurements must first be corrected for the vertical profile of precipitation (VPP) which includes changes in size, phase and fall speed of hydrometeors. Because vertical profile corrections are typically based on and applied to the radar reflectivity measurements, this technique is called the vertical profile of reflectivity (VPR) correction.

The existing VPR correction methods can be subdivided into three broad types (Germann, 2000; Zhang and Qi, 2010) which are based on climatology, spatio-temporal averages and modelled VPRs. Climatological VPRs are based on radar data averaged over long time periods (days, seasons, years) and over a certain spatial area (radar volume or well-visible regions) (Joss and Pittini, 1991; Joss and Lee, 1995). Advantages of these type of VPRs is that, once calculated, they are computationally

inexpensive, based on actual radar data and thus always available. However, the climatological VPR assumes both spatial and temporal homogeneity, while in reality important variations may occur, for example depending on whether precipitation is of stratiform or convective type. Operationally, the climatological VPR is often used as a default VPR in case the real-time VPR is unavailable.

Compared to the climatological VPRs, spatio-temporally averaged VPRs can better capture the temporal variations in re-

flectivity since these are based on a few volume scans only and regularly updated. They therefore also remain computationally inexpensive, and among the few countries who correct for VPR in the operational processing, several are using some version of spatio-temporally averaged profiles (Koistinen, 1991; Joss and Lee, 1995; Germann and Joss, 2002). In the MeteoSwiss operational network profiles are calculated in well-visible (clutter free) regions around the radar at the meso-beta scale (integrated over a few hours and within a range of 70 km). For the aggregation in space, the polar pixels are weighted by the area of the

corresponding pulse volume. For the aggregation in time, a precipitation- and volume weighted exponentially decaying function is applied, using the profile of the most recent precipitation event as a first guess. The aggregation time is variable, such that it is more regularly updated for widespread rainfall while longer time aggregation is performed for intermittent rainfall in order to smooth the profile (Germann and Joss, 2002). This technique has the advantage of always providing a vertical profile, with smooth transitions between subsequent radar scans and within reasonable processing times.





An alternative method to better account for spatial variability of profiles is to use a VPR model to obtain a profile at each location (e.g. Kitchen et al. 1994 and Kirstetter et al. 2013). These can be determined using a set of physically based parameters in order to remain computationally inexpensive. The UK Met office, for example, uses parametrisations for the melting layer (NWP model freezing level height), orographic growth (Hill, 1983) and the cloud top height (satellite infrared imagery) (Harrison et al., 2000). The parametrised vertical profile is then weighed by the reflectivity factor measured just beneath the bright band (Harrison et al., 2000). A suggested extension of this method includes the use of LDR measurements to identify whether a stratiform or a convective type of profile should be applied (Sandford et al., 2017). More recently, Le Bastard et al. (2019) have proposed an approach based on simulated VPRs from the NWP model AROME-NWC which are matched with the most similar observed VPR by the radar. Although the model based VPRs have the advantage of providing a profile at each radar pixel, the disadvantages include its dependency on the availability and quality of information from external sources and in some cases, on a-priori assumptions on the shape of the VPR (i.e. the slope of the VPR in the solid layer).

Therefore, the majority of existing operational VPR correction schemes assume spatio-temporal homogeneity and rely heavily on the reflectivity measurements. Due to the spatial variability of precipitation microphysics, of temperature and humidity profiles as well as the growth and decay processes (Matsuo and Sasyo, 1981; Fabry and Zawadzki, 1995; Bell, 2000; Roe, 2005; Stoelinga et al., 2013), VPP and thus VPR profiles may be expected to vary considerably in space and time, especially in an orographic context (Boodoo et al., 2010; Campbell and Steenburgh, 2014).

The aim of this study is to propose a more localised vertical profile correction technique using machine learning (ML) and information on hydrometeor proportions to predict the vertical change in reflectivity, referred to as *growth and decay* (GD). To the best of our knowledge, in the literature there is no ML technique for the investigation of the vertical structure of precipitation that is mature enough for operational implementation, and therefore an important part of this work is to present a proof of concept for the use of ML in this context. Artificial Neural Networks (ANN) comprise a class of ML methods which are well-established in the geo- and environmental sciences and are also used for this study.

The selection of precipitation events, preparation and extraction of the data will be addressed in Sect. 2, while more details on the set up and the training of the ANN are given in Sect. 3. The main objective of this work will be addressed through the following steps: first, the ANN is used to learn about the contribution of hydrometeors to potentially improve radar-based QPE in Switzerland, then information at lower height levels is gradually removed in order to test the ANN's ability to extrapolate the radar measurements to the ground level. Finally, the ANN predictions will be evaluated and compared with the current operational VPR correction technique. Sect. 4 presents the exploratory data analysis, the results of the machine learning predictions and their verification, while Sect. 5 puts the contribution of this study into a broader perspective.

## 2 The vertical cone database

The data used in this study are extracted from high-resolution volumetric radar data acquired by the Albis radar located at ∼900 m a.s.l on the Swiss plateau. The radar has good visibility from the South West to the East and some regions with residual





ground clutter in the South due to the presence of the first Alpine slopes (Fig. 1). Of the five Swiss radars, the Albis radar is situated at the lowest altitude and can thus provide measurements of the lowest parts of the atmosphere. Moreover, the Albis radar has been stably producing high quality data over the past years, allowing the ANN to be trained with the best available quality data.

5                                                        [FIGURE 1 ABOUT HERE]

## 2.1 Radar data pre-processing

Like the other radars in the Swiss operational network, the Albis radar performs 20 plan position indicator (PPI) scans within five minutes at elevations ranging from -0.2° to 40° (Germann et al., 2015) with an interleaved sweep pattern. For this study, the high resolution data with a range resolution of 83 m (corresponding to 12 bins every 1 km) was used.

The processing of the radar data was performed using the Python-based open source software Pyrad (Figueras i Ventura et al., 2017), which was developed at MeteoSwiss as an extension to Py-ART (Helmus and Collis, 2016). The signal-to-noise ratio (SNR) of the horizontal channel was calculated based on the estimated receiver noise at high-elevation angles (40° or 35°). Subsequently the SNR, and the ratio of the receiver noise in the horizontal and vertical channels were used to filter out the noise before computing the co-polar correlation coefficient ($\rho_{HV}$) (Gourley et al., 2006). Clutter was identified and

removed using a filter based on textures of reflectivity factor at horizontal polarisation ($Z_H$), differential reflectivity ($Z_{DR}$), copolar cross correlation coefficient ($\rho_{HV}$), total differential phase shift ($\psi_{dp}$) and the value of $\rho_{HV}$. Range gates with an SNR below the threshold of 10 dB were filtered out before performing a double window moving median filter on $\psi_{dp}$. The filtered differential phase ($\phi_{dp}$) was then used to estimate the specific attenuation ($A_h$) using the ZPHI algorithm (Testud et al., 2000) in order to correct for attenuation induced by precipitation, and to derive the specific differential phase shift on propagation

($K_{dp}$) using the method described by Vulpiani et al. (2012).

The filtered and corrected polarimetric variables were then used as input for the semi-supervised hydrometeor classification scheme developed by Besic et al. (2016, 2018). This method uses $Z_H$ (-10 - 60 dBZ, influenced by particle concentration, size and density), $Z_{DR}$ (-1.5 - 5 dB, influenced by particle shape, orientation and density), $K_{dp}$ (-0.5 - 5 deg km$^{-1}$, influenced by particle concentration and shape), $\rho_{hv}$ (0.7 - 1, influenced by particle homogeneity) and a liquid/melting/ice phase indicator

to distinguish nine classes of hydrometeor types. These classes consist of: aggregates (AG), ice crystals (CR), light rain (LR), rimed particles (RP), rain (RN), vertically-oriented ice crystals (VI), wet snow (WS), melting hail (MH), ice hail high-density graupel (IH/HDG) and no classification (No valid radar data - NC). Within the context of this study, light rain and rain were aggregated into a single rain class and vertical ice was aggregated into the crystals class.

## 2.2 Vertical cone definition

30                                                       [FIGURE 2 ABOUT HERE]

Because this study aims to propose a more localised vertical profile correction technique, the filtered and corrected volumetric radar data need to be sampled at scales small enough to capture the spatio-temporal variability and large enough to give a



robust estimate of the vertical profile at each location. Rather than extracting a vertical column of a certain size, vertical cones such as those illustrated in Fig. 2 were extracted. The motivation for this is twofold: on the one hand, it follows the assumption that precipitation observed at a certain point of interest $x, y$ on the grid (Fig. 1) may have originated from a much wider region aloft, and, on the other, it accounts for the decrease in the number of measurements at higher altitudes by increasing the

sampling size of the cone.

As described in Roe (2005), the terminal fall speed of hydrometeors varies between 1 m s$^{-1}$ for snow and 10 m s$^{-1}$ for heavy rain (excluding hail stones). This implies that when taking into account a range of horizontal wind speeds from 5 to 30 m s$^{-1}$, a hydrometeor originating at 3 km altitude may get advected anywhere between 1.5 and 90 km before reaching the surface (Roe, 2005).

However, within the context of this study, the choice of the cone size, i.e. the diameter of its base and top, is related to grid spacing: as is shown later, cone sizes were chosen which do not overlap or touch at the base and partly overlap at the top. In addition, the cone size should be chosen such that there is a sufficient number of samples at each altitude and at all distances from the radar.

Considering that for the lowest elevation angle the 1° beam width diameter exceeds 1 km at distances further than 60 km, it

was estimated that a 500 m vertical resolution was the highest possible. Given the 500 m vertical resolution and distances of >60 km from the radar, it was also estimated that cones with a 4 km radius at the base and a 10 km radius at the top, would have sufficient samples at each height level (Table 1).

For this exploratory study, the dimensions of the cone have been chosen for practical reasons (enough samples at each height level in each of the cones up to a 60 km distance from the radar). The possibility of using higher vertical resolutions

will be the subject of a follow-up study. In order to preserve some spatio-temporal consistency and to further increase the number of samples at each height level, the cones at each location were aggregated over up to 30 minutes (6 previous scans). This spatio-temporal averaging has a solid physical-meteorological motivation (Germann and Joss, 2002). The 10 km and 30 minute scales have been selected because it is expected that at these scales the lower part of the VPP can be related to the VPP and hydrometeor proportions aloft. Such a relation is not reasonable and difficult to observe at higher resolutions.

25                               [TABLE 1 ABOUT HERE]

## 2.3   Extraction of variables

From each cone, the input data to train the ANN (i.e. the horizontal reflectivity factor $Z_H$ and the hydrometeor class (HC) proportions) were extracted within height level bands of 500 m at levels ranging from 1500 m to 10 km altitude. The lowest available level (1000 m a.s.l.) was considered the "ground" reference. For $Z_H$ the average reflectivity (in linear units) within

each height band was calculated, including measures of dispersion and location (standard deviation, percentiles 16, 25, 50, 75 and 84). For the hydrometeor classes the number of pixels pertaining to each class was counted and these were transformed into proportions after temporal averaging. The temporal averaging was performed as a last step so that each individual cone could still be stored and examined and such that experiments could be performed with shorter temporal averaging, if needed.





The target value which the ANN was required to predict was chosen to be the vertical multiplicative growth and decay (GD) of precipitation:

$$y = GD_{h-1\,km} = 10 * \log_{10}\left(\frac{Z_{h\,km}}{Z_{1\,km}}\right) \text{[dB]} \tag{1}$$

which was derived after the temporal averaging and for each height level $h$. The multiplicative definition of growth and decay

was also used by Foresti et al. (2019) but in the nowcasting context, more precisely to predict the horizontal (instead of vertical) change of radar precipitation when advected downstream (Lagrangian frame).

The choice for trying to predict growth and decay was based on the accurate definition of the learning problem at hand; here we are interested in predicting the vertical change in reflectivity between the lowest visible height level and the ground, rather then in predicting the absolute reflectivity value at the ground level. The reason for this is twofold. Firstly, this is more

comparable to the operational VPR method, which is also based on a relative correction factor. Secondly, it simplifies the categorisation of the physical processes into enhancement and dissipation of precipitation.

## 2.4 Selection of precipitation events and cone locations

Since the extraction and calculation of the variables inside the cones is a costly operation in terms of computing time, there is some trade-off between the number of cones to be extracted per scan and the number of precipitation events considered.

For this study, 30 precipitation events over the course of three years (2016, 2017 and 2018) and covering a broad range of microphysical situations were selected. Within each event, one scan per hour was selected, and for each scan the 6 previous scans (equalling 30 minutes in time) were also extracted. The temporal spacing between the extracted scans was adopted in order to reduce the correlation between successive temporally averaged cones and allowed to further limit processing time. Details of the events are given in Table 2. Due to the higher frequency and duration of stratiform events these are slightly

over-represented in the data.

[TABLE 2 ABOUT HERE]

Finally, for each single cone location the vertical profiles of reflectivity were calculated and averaged over the entire data set. This was done to exclude cones with consistently missing data at certain height levels related to the radar scan geometry; this is different from the information in Table 1 which shows the theoretical median number of pixels within cones at different ranges

from the radar. Indeed, based on this analysis the cones at 85 locations, usually situated far from the radar were removed. The reasons for removing these data points were that the ANN cannot handle predictors with missing data and that these cones resulted in aberrant GD values in the observation data set. For the same reasons, cones which were less than 10% filled in the bottom 4000 metres of the cone were also removed, such that a total of 17123 cones remained for input in the ANN models.





## 3  Neural network and experimental setup

Machine learning algorithms are tools which, compared to traditional statistical data models, are fully non-parametric and designed to solve regression tasks in high-dimensional input spaces. This means that ML aims to obtain the best possible performance without making strong assumptions about the distributions of or dependency between the variables (see e.g.

Kanevski et al. 2009). Artificial Neural Networks (ANN) are a type of ML which are particularly well adapted to treating multi-dimensional input data and resolving non-linear problems. In this study, we employed a classical back-propagation feed-forward MultiLayer Perceptron (MLP) for non-linear regression. The Python Scikit-learn (Pedregosa et al., 2011) library was used for the experiments.

MLP models are typically composed of one input layer, one or more hidden layers and one output layer (Fig. 3). The neurons

in the input layer correspond to the input variables (predictors) and the neurons in the output layer to the target variables (predictands). Each neuron is connected to the other neurons of the network with synaptic weights $\boldsymbol{w_i}$ (which can take positive or negative values). The hidden neurons integrate the input signals from the ones in the previous layer and apply a bounded, non-linear sigmoid activation function (typically logistic or hyperbolic tangent) before passing the output $o_i$ to the next layer. The combined effect of the hidden layer is that it performs a non-linear transformation on the weighted linear summation of

the values coming from the input neurons. Using a design with less neurons in the hidden layer than the input layer allows performing data compression. Conversely, when the number of hidden neurons is larger it allows representing the input data in a higher dimensional space to better shatter (discriminate) the data.

[FIGURE 3 ABOUT HERE]

The training of the MLP is then achieved by performing an iterative gradient-based optimisation of the network weights $\boldsymbol{w}$

to minimise the mean squared error loss function between the predicted and the target values in the output neuron:

$$E_{MSE}(\boldsymbol{w}) = \frac{1}{N} \sum_{i=1}^{N} \left( y_i^{obs} - y_i^{pred}(\boldsymbol{x}, \boldsymbol{w}) \right)^2 \tag{2}$$

Provided that there are a sufficient number of neurons and a non-constant, non-decreasing activation function at each hidden neuron, an ANN can model any non-linear function with the desired precision (Cybenko, 1989; Hornik et al., 1989). This, however, means that a too complex network architecture may lead to overfitting of the data. On the other hand, a too simple

network, or in the case the training is trapped in a local minimum of the error function, could cause underfitting. Both cases lead to the inability of the ANN to generalise the learnt patterns. In this paper, overfitting is addressed by early stopping, while stochastic gradient descent is used to avoid getting trapped in local minima.

Thus, the main hyper-parameters of MLP models are the number of hidden layers, the number of hidden neurons within these layers, the type of activation function, the learning rate for the update of the weights, and the criteria for early stopping.

The training of the MLP, the selection of hyper-parameters as well as the evaluation of its performance on completely independent data is achieved by splitting the data. In this paper, the data set is split into training (60%), validation (20%) and testing



(20%) data sets. While the model is trained with the training data set to find the appropriate weights to minimise the training error, it is simultaneously applied to the validation data set in order to evaluate the predictive performance of the MLP during training. The training error will continue to decrease, however as soon as the model starts overfitting the training data, the validation error will start to increase. The set of weights with the lowest training and validation error is retained as the best

model.

The testing data set is used as a completely impartial data set to evaluate the models. In this study, all three data sets contain cones from separate precipitation events with as a constraint that each data set includes at least one event from every season.

Two main types of ANN models are trained and tested within the scope of this study: one model is trained using only values of average reflectivity at each height level ("dBZ-only model"), while the second is trained with the average values

of reflectivity and HC proportions of rain, wet snow, rimed particles, aggregates and crystals at each height level ("HC + dBZ-model"). Occurrences of hail and melting hail are so rare in the considered data set that they are not considered as input variables. The goal is to understand how relevant is the information on hydrometeor types, in addition to reflectivity, to extrapolate precipitation to the reference level.

For each ANN type, separate models starting at different height levels are trained (i.e. starting at 1500 m a.s.l. and upwards,

starting at 2000 m a.s.l. and upwards etc.). Because the number of input neurons is equal to the number of variables × the number of altitudes, the dBZ-only model for example has 20 neurons in the input layer for the ANN starting at 1500 m a.s.l, 19 neurons for the ANN starting at 2000 m a.s.l. and so forth. For the HC + dBZ-model these numbers are 110 and 104 respectively.

## 4   Exploratory data analysis and results

The goal of exploratory data analysis (EDA) is to better understand the dataset and extract some useful information before applying ML. The efficiency and accuracy of ML methods depend critically on the quality and quantity of the data, as well as the existence of statistical relations between variables (patterns). EDA may therefore be useful for the identification of relevant variables, clearly observable patterns, or outliers in the data set.

### 4.1   Vertical profiles of hydrometeor proportions

In order to evaluate the ability of the vertical cones to robustly describe the vertical structure of precipitation and as a part of the EDA, the vertical profiles of hydrometeor proportions, reflectivity and GD have been plotted. Figure 4 shows the medians and quartiles of the vertical profiles for different events calculated over the entire spatial domain and for a single 30 minute time step. The profiles of hydrometeor proportions show some distinctly different features for events with snow at the ground, stratiform and convective precipitation. For the event with snow reaching the ground level (left panel Fig. 4) high proportions

of aggregates and rimed particles can be observed at lower altitudes, while above 5500 m a.s.l. only crystals are present. For the stratiform event, crystals also dominate at altitude levels above 5500 m a.s.l., but a distinct increase in wet snow around 2000 m a.s.l. indicates the presence of a melting layer, with rain dominating at the lower altitude levels. For the convective event,



rain dominates up until 3000 m a.s.l. and an important proportion of crystals, aggregates and rimed particles can be observed until at least 10 km altitude.

The vertical profiles of hydrometeor proportions in Fig. 4 indicate that different event types can be distinguished by the HC proportions from the cones and that some types of events may benefit more than others from the information aloft.

[FIGURE 4 ABOUT HERE]

## 4.2 ANN predictions of growth and decay

The 2D histogram plots in Fig. 5 show the observed (x-axes) and predicted (y-axes) GD for the HC + dBZ-models (top row) and dBZ-only (bottom row) trained with data from different altitudes and upwards (columns). As extrapolations are made from increasingly higher altitudes, both the observations and the predictions move towards growth, as can be expected. The

colouring of the plots indicates the point density, and while the blue points are single observations and thus show more scatter, the areas with high point density (red/orange) fall better along the identity line for the HC + dBZ-model than for the dBZ-only model. Especially for predicting GD from 2500 m a.s.l and aloft, the dBZ-only models seem to have difficulty with predicting growth values higher than 10 dB.

[FIGURE 5 ABOUT HERE]

The 2D histogram plots in Fig. 5 are summarised in Fig. 6, which shows the Root Mean Squared Error (RMSE) and Pearson correlation coefficient ($\rho$) between the observed and predicted GD in dB units. It can be observed that the model with HC proportions performs consistently better than the model without HC proportions, i.e. at equal RMSE values, the HC + dBZ model can predict from altitudes between 500 and 1000 metres higher than the dBZ only model. The RMSE for both model types seems to level off for ANNs trained with data starting from 4000 m a.s.l. and aloft, highlighting the upper limit of

extrapolation (experiments performed up to 10 km a.s.l. show that RMSE remains constant also at these altitudes). At the lowest height level (1500 m a.s.l.) the dBZ-only model and the HC + dBZ-model give similar errors. A possible explanation for this is that for the prediction of GD between 1500 m a.s.l. and the reference level (at 1000 m a.s.l.), the average reflectivity at 1500 m a.s.l. is the dominant variable also for the HC + dBZ-model. The differences between the models and observations at this and subsequent height levels is analysed in more detail in the following section.

[FIGURE 6 ABOUT HERE]

### 4.2.1 2D matrices of growth and decay

In this section, we want to explore the growth and decay patterns in the space of predictors to visually verify whether the ANN properly learnt the observed data dependencies.

The matrices in Fig. 7 show the binned averages of GD based on combinations of HC proportions and average reflectivity

values at the lowest height levels (1500 m a.s.l. and 2000 m a.s.l.) for the observed data (left column), the HC + dBZ-model (middle column) and the dBZ-only model (right column). This is similar to computing a 2D histogram, but instead of counting





the number of samples it evaluates the average value. The GD values were calculated based on the reflectivity at that height level and the 1000 m a.s.l. reference level. The plots allow to distinguish different GD patterns for different combinations of variables. Overall, higher average reflectivity values at any altitude $h$ lead to observed decay between that altitude $h$ and the ground reference and, inversely, lower reflectivity values at altitude $h$ lead to more observed growth. This is also reflected by

the models. However, more specific patterns can also be observed, such as the pronounced growth values for cones with low to moderate reflectivity values at altitude $h$ and high proportions of aggregates or any presence of rimed particles at that same altitude $h$. While these patterns are also visible in the HC + dBZ-model output, the dBZ-only model is unable to reproduce them.

In order to evaluate how well the models reproduce the observed GD patterns, Figs. 8 and 9 shows the 2D error matrices. For

each of the combinations of variables the binned 2D averages for the observations was subtracted from the binned 2D averages of the model outputs such that positive values in Figs. 8 and 9 show model overestimation and negative values correspond to model underestimation.

[FIGURE 7 ABOUT HERE]

Overall, the errors for the dBZ-only model have a greater amplitude than the errors for the HC + dBZ-model (Figs. 8 and

9). Most notable is the overestimation by the dBZ-only model for cases where crystals are present (col. 3). For example, the RMSE for combinations of crystals and meandBZ at 1500 m a.s.l. for the HC + dBZ-model is 3.05 dB while for the dBZ-only model it is 4.64 dB (row 1, col. 3). The dBZ-only model also particularly underestimates for cases with rimed particles and high proportions of aggregates (row 2, col. 4). The HC + dBZ-model also underestimates slightly the cases with rimed particles, though only when predicting from 1500 m altitude levels, and less so than the dBZ-only model. More precisely, combinations

of rimed particles and meandBZ at 1500 m a.s.l. result in an RMSE of 2.23 dB for the HC + dBZ-model and 2.90 dB for the dBZ-only model (row 1, col. 4). Overall, the HC + dBZ-only model shows less patterns in the errors, which indicates that the ANN model better separated the signal from the error. The largest deviations are located at the edges of the distributions and are thus more likely related to outliers in the observations. This is a good behaviour and highlights the ability of the ANN to remain robust to outliers (no overfitting).

[FIGURE 8 ABOUT HERE]

[FIGURE 9 ABOUT HERE]

Figures 10 and 11 are the same as Fig. 5, but, instead of comparing the instantaneous predicted and observed growth and decay values, it compares their binned averages of Figs. 7 and Figs. 8 and 9. As expected, the correspondence of the average values is much better than the one of the instantaneous ones.

The numbers mentioned in the discussion below are relative to the 1500 m level, but the statements are also valid at 2000 m. For the HC + dBZ-model the observed and predicted GD for combinations of average reflectivity and any hydrometeor class show good agreement with high regression slopes ($\beta < 0.54$) and correlation coefficients ($>0.77$). For combinations of hydrometeor classes, and especially rain + aggregates or rain + rimed particles the performance of the HC + dBZ-model is not





as good (correlations drop to 0.63 and 0.58 respectively), which could be related to the more complicated and less frequent nature of situations with these combinations of hydrometeor classes. For the dBZ-only model, the agreement between observed and predicted GD for combinations of average reflectivity and hydrometeor class is much lower than for the HC + dBZ-model. For combinations of hydrometeor classes the dBZ-only model gives similar predictions for the whole range of observed GD

values. This demonstrates the added value of using polarimetric information (through HC proportions) to using the reflectivity data alone.

[FIGURE 10 ABOUT HERE]

[FIGURE 11 ABOUT HERE]

### 4.2.2 Comparison ANN predictions with traditional methods

The comparison between the ANN model outputs and traditional VPR correction techniques is made by adding the predicted GD to the lowest reflectivity measurement and comparing the predicted reflectivity at the ground level with the observed reflectivity:

$$RMSE = \frac{1}{N} \sum_{i=1}^{N} \left( dBZ_i^{obs} - dBZ_i^{pred} \right)^2 \tag{3}$$

where $dBZ_{pred}$ for the ANN models is obtained by:

$$dBZ_{pred} = dBZ_h + GD_{h-1\,km} \tag{4}$$

The traditional models considered are:

1) performing no correction, i.e. assuming *vertical persistence* by taking the lowest available reflectivity measurement:

$$dBZ_{pred} = dBZ_h \tag{5}$$

2) applying a constant gradient of 1.5 dBZ / km:

$$dBZ_{pred} = dBZ_h + \Delta h * 1.5 \tag{6}$$

3) the *meso-beta* profile correction factor which is calculated operationally. For each altitude $h$ the correction factor is extracted from the profiles and applied to the average reflectivity value at altitude $h$ from the cone. More details on the calculation of the correction factor can be found in Germann and Joss (2002).

The RMSE for each of these VPR models as well as the dBZ-only and HC + dBZ models were calculated over 10 combina-

tions of completely independent test datasets and are given in Fig. 12.

[FIGURE 12 ABOUT HERE]





For predictions made from altitudes up to approximately 2500 m a.s.l., assuming a constant gradient gives the worst results, while assuming vertical persistence may be feasible at lower elevations but results in large errors when the lowest visible elevations are higher than 2000 m a.s.l.

The operational meso-beta profile is extracted from the well-visible regions close to the Albis radar and gives a correction
factor with respect to the reference altitude used operationally and which is set to 1500 m a.s.l.. Since the meso-beta profiles have no information at 1000 m a.s.l. altitude, the initial error for predictions from 1500 m a.s.l. to 1000 m a.s.l. is approximately 1.6 dB higher than for the ANN models. Because the meso-beta profile correction factors are calculated such as to obtain a more or less constant rain rate in the vertical, the RMSE of the meso-beta profile also remains quite constant up to 4000 m a.s.l. The increase in RMSE at higher altitudes is probably because the required correction exceeds the maximum threshold for the
operational correction factor. Compared to the traditional models the ANN models show substantial improvement, especially when using the HC proportions.

Finally, as already observed in Fig. 6 the error levels off around 4000 m a.s.l. for both ANN models. This may be partly explained by some over representation of stratiform events in the dataset, which are less developed in the vertical so that the models have very little information available at these altitudes.

15                                         [FIGURE 13 ABOUT HERE]

Figure 13 shows the models compared in Taylor diagrams. These diagrams show the Pearson correlation coefficient on the azimuthal angle, the centered RMS error (green contours) and the standard deviation of the models as the radial distance from the origin. The observed data is indicated by a star. For the HC + dBZ model predicting from 2500 m a.s.l. (first row, third column, blue hexagon) the correlation coefficient is about 0.72, the RMSE about 4.6 dB and the standard deviation about 5.8
dB. Taylor diagrams are able to show that even if models may have a similar RMSE, one model may better correlate with the observations or have a similar standard deviation as the observations. For the models depicted in Fig. 13 we can observe that while the traditional models have increasingly higher RMSEs and smaller correlations when predicting from higher altitudes, the ANNs tend to have similar correlations but smaller standard deviations. This last observation is typical for ML methods and is due to the mean square error minimisation (Eq. 2). Because the RMSE is equal to the square of the bias plus variance
(plus the irreducible error), the reduction in RMSE also reduces the variance in the model predictions.

## 5   Conclusions

The aim of this study was to propose a more localised vertical profile (VPR) correction technique by making use of machine learning (ML) and by exploiting polarimetric radar information through the use of hydrometeor types and their proportions. An important part of the work consisted of establishing the foundations for the use of ML for the investigation of the vertical
structure of precipitation.

Vertical cones were extracted on a regular grid up to 60 km distances and in the well visible regions of the Albis radar. The cones were divided into height levels from 1500 m a.s.l. up to 10 km a.s.l. with a 500 m vertical resolution. For each



500 m height level band, the average reflectivity values and hydrometeor proportions were calculated and used as inputs for the Artificial Neural Network (ANN) model. The ANN model was chosen because it offers a smooth estimation of non-linear functions in high-dimensional spaces. The target value (predictand) for the ANN model was the vertical change in reflectivity (or growth and decay, GD) between each height level and the reference "ground" level (1000 m a.s.l.). A total of 30 precipitation

events were randomly split into training, validation and testing datasets, each containing data from separate events. The ANN was then trained, calibrated and evaluated with completely independent test datasets.

Exploratory data analysis (EDA) of the vertical cone data allowed to further filter the dataset and to exclude cones with consistently missing values at certain height levels due to the geometrical constraints related to the radar scan strategy. EDA also allowed to verify that the cones could successfully capture the vertical structure and hydrometeor proportions of the various

types of precipitation events.

In order to evaluate the potential of operational information on hydrometeor classes (HC) to improve quantitative precipitation estimates (QPE) in Switzerland, two main types of ANN were trained: one using only the average reflectivity values at different height levels (dBZ only-model) and one using the average reflectivity values and hydrometeor proportions at each height level (HC + dBZ-model). The ability of each model to extrapolate the radar measurements from higher altitudes to the

ground level was then assessed by progressively removing information at the lower height levels of the vertical cones and re-training the ANNs. It was found that, for equal values of RMSE, the HC + dBZ-model could predict from altitudes between 500 and 1000 metres higher than the dBZ only-model. A more in-depth analysis of the GD patterns as a function of hydrometeor types, indicated that the dBZ-only model overestimated (underestimated) GD especially in cases where crystals (aggregates) were present.

Finally, the ANN models were compared to traditional VPR correction techniques by adding the ANN predicted GD to the lowest observed reflectivity value. The other approaches considered were: vertical persistence of reflectivity, a constant gradient and the operational meso-beta profile which was extracted for each time step and applied to the cone data. It was found that both ANN models performed better at all height levels than the traditional VPR correction techniques. The higher error observed for the meso-beta profile correction technique may have been partly caused by the fact that this method uses

1500 m a.s.l. as the reference level, and not 1000 m a.s.l. The performance for both the dBZ only-model and the HC + dBZ-model levels off above 4000 m a.s.l., suggesting that the models have little or no predictive skill above this altitude. Finally, it could be observed that for traditional models the RMSE increases for predictions made from increasingly higher altitude levels while the correlation of the predictions with the observations decreases. For the ANN models, the RMSE increases less, but the standard deviation of the predictions decreases.

Future work could include a sensitivity analysis of the contributions of the input variables, as this would allow to remove redundant predictors and so further simplify the models. Similarly, an evaluation of the influence of the geometry and spacing of the cones on the final result (within the aforementioned constraints related to processing time, the resolution of the radar measurements and visibility at lower height levels) would allow to further improve the results of this method. Finally, the here presented cone extraction and evaluation of the method has been performed entirely on data from the Albis radar. As such,

it assumes that the physical processes and the relationship between these processes are the same in other locations (such as





above mountainous valleys). It would be valuable to perform and compare the vertical cone correction method on one of the high altitude radars in Switzerland. However, this would require installing a vertically pointing radar at the bottom of the valley to acquire the reference observations below the lowest operational C-band radar measurements (currently at around 3000 m a.s.l.).

5    The presented machine learning predictions are deterministic and are consequently affected by conditional biases, which includes a loss of variance of the predictions (w.r.t. the observations) due to the error minimization principle (e.g. Foresti et al., 2019). Future research could consider the combination of probabilistic machine learning models and stochastic simulation to generate radar rainfall ensembles (Jordan et al., 2003; Bowler, N. E., Arribas, A., Mylne, K., Robertson, K., and Neare, 2007; Kirstetter et al., 2015; Foresti et al., 2019).

10    The approach proposed in this study takes advantage of the capability of ML techniques to learn complex non-linear relationships between polarimetric radar variables (represented by the HC proportions) along the vertical column. It demonstrates their potential to improve the extrapolation of high altitude radar observations to lower levels, which is a relevant step for the improvement of polarimetric radar QPE in complex terrain.

15    *Author contributions.*  All authors contributed to the development of the concept and methodology presented in the paper as well as the interpretation of the results. FH and LF performed the analyses. FH, with contributions from all authors, prepared the manuscript.

*Competing interests.*  The authors declare that they have no conflict of interest.





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





## List of Figures









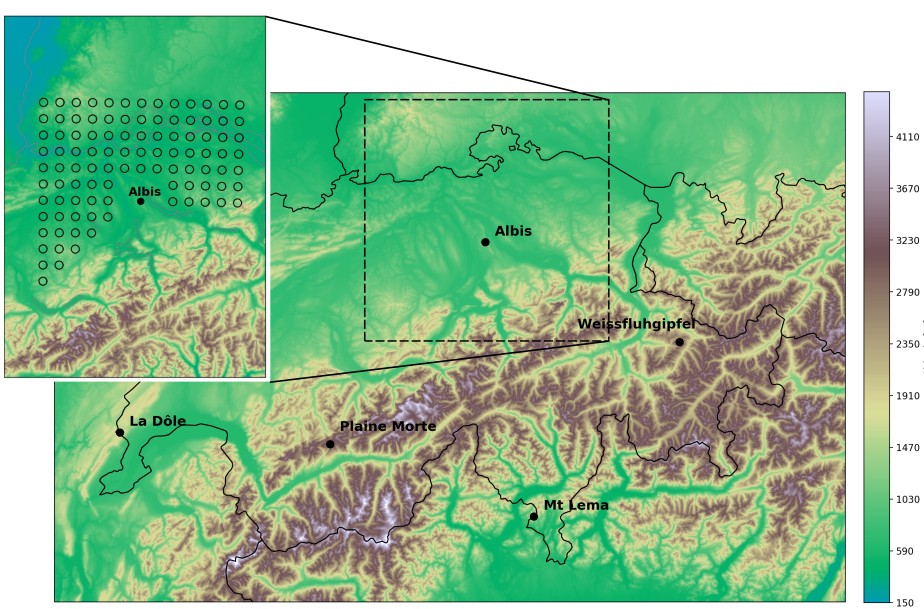

**Figure 1.** Location of the Albis radar within Switzerland overlaid on the digital elevation model (DEM, Jarvis et al., 2008) and the regular grid for the vertical cone extraction.





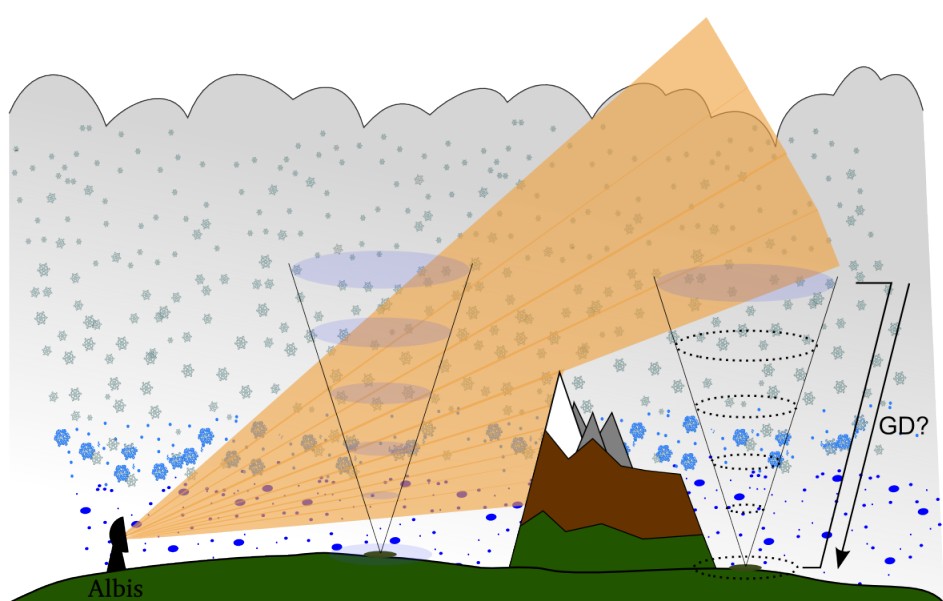

**Figure 2.** Cones are extracted in well-visible regions of the radar and used to train the ANN. The trained ANN model can then be used to extrapolate measurements from aloft to the ground level during other precipitation events or in regions with reduced visibility.



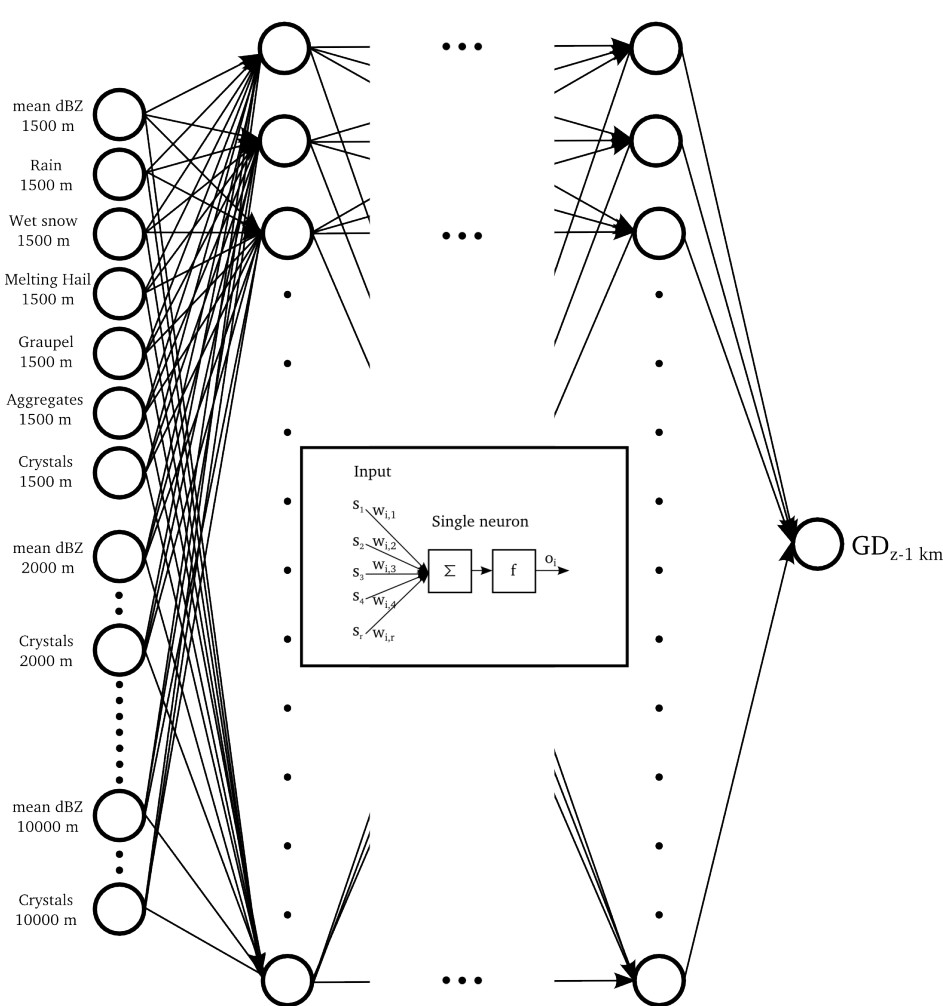

**Figure 3.** Example of a multi-input single-output MLP model and set-up as used in this study. The number of input neurons $M$ equals the number of $V$ predictors (reflectivity, HC proportions) $\times$ $H$ height levels used.



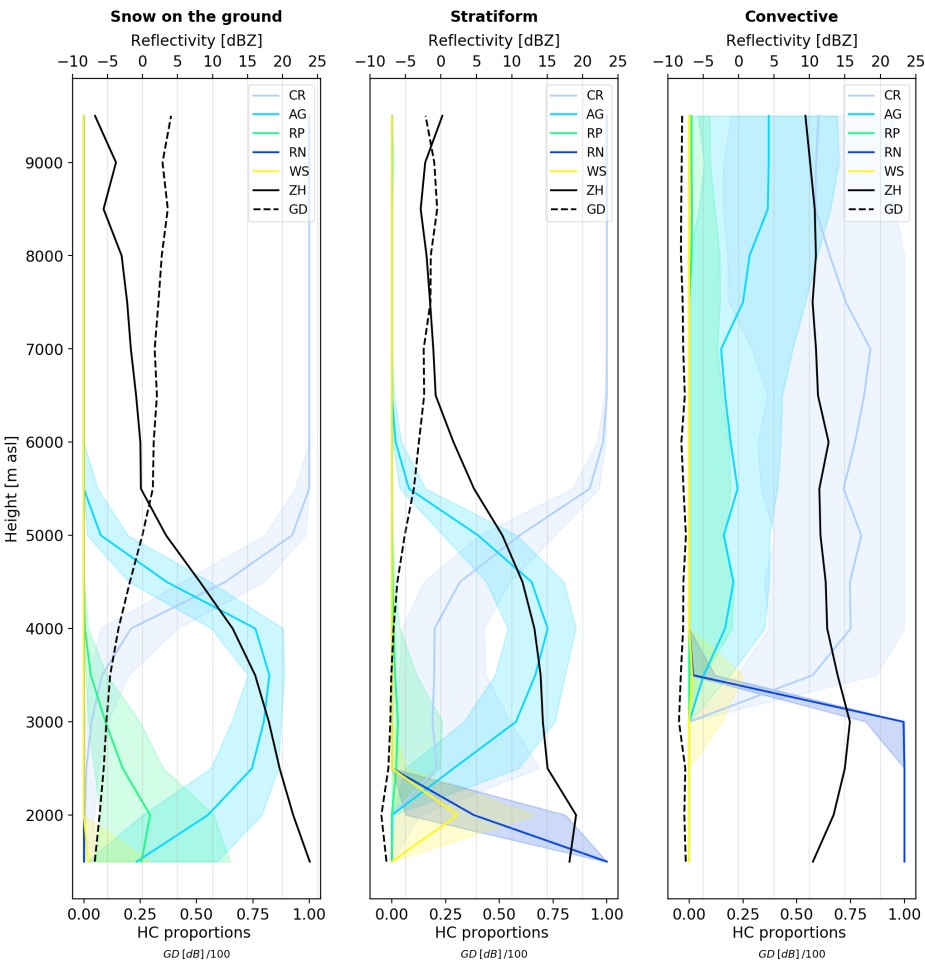

**Figure 4.** Examples of vertical profiles of reflectiviy, GD and hydrometeor proportions for three different events. Thick lines show the median values and shading the quartiles calculated over the entire spatial domain for a single 30 minute time step.

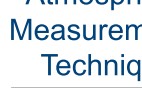
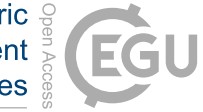

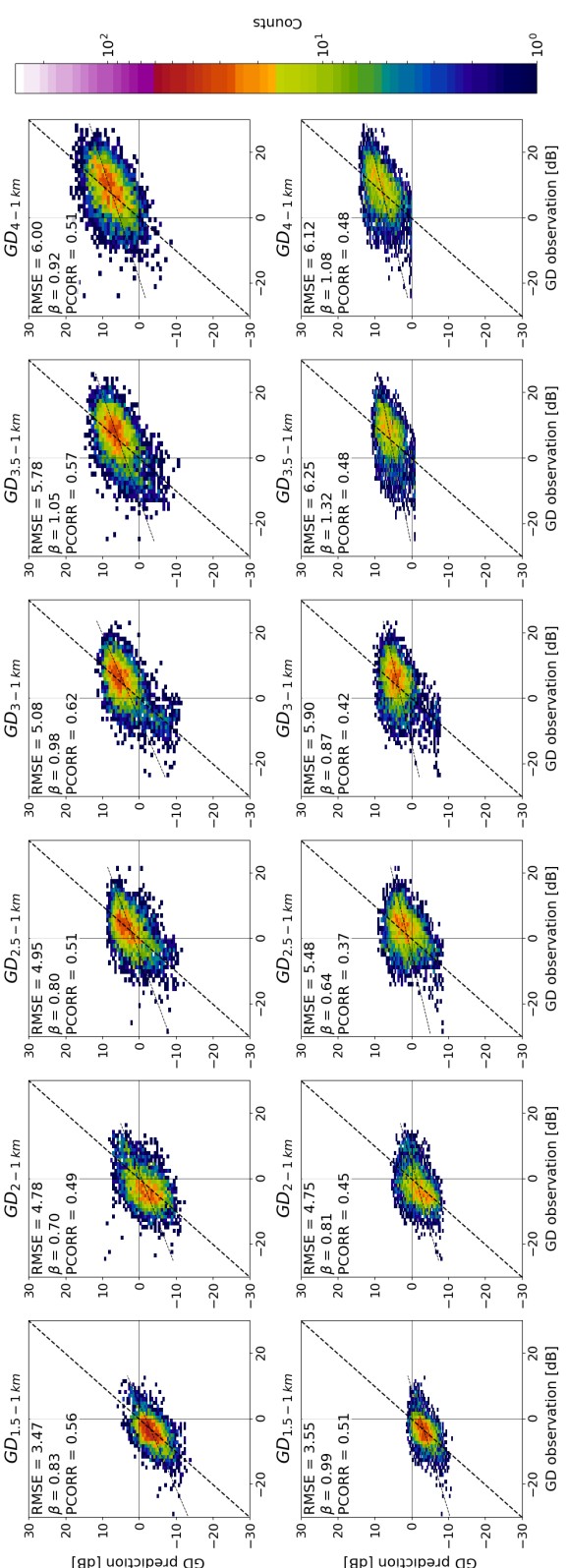

**Figure 5.** 2D histograms for observed (x-axes) and predicted (y-axes) GD for the HC + dBZ-models (top row) and dBZ-only models (bottom row) trained with data starting from higher altitude levels and aloft (columns). The number of points in each bin is indicated in colour.





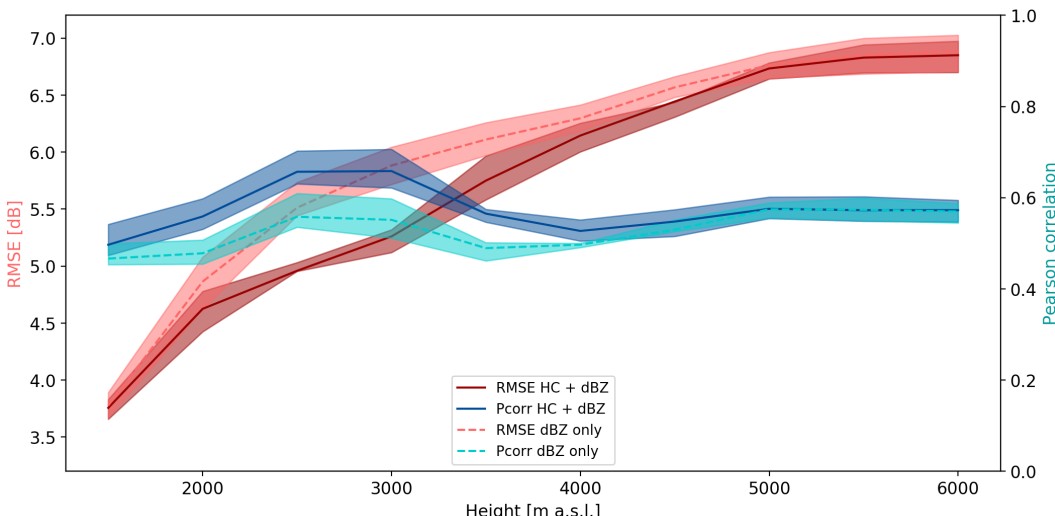

**Figure 6.** RMSE and Pearson correlation coefficient for dBZ only-models and HC + dBZ-models trained with data starting from increasing altitude levels. The thick lines indicate the average values and the shading the quartiles calculated over ten model runs using different combinations of events for the training, validation and testing datasets.







**Figure 7.** 2D growth and decay matrices for 1500 m a.s.l. (top row) and 2000 m a.s.l. (bottom row) for observed data (left), HC + dBZ-model (center) and dBZ-only model (right). The colour of each bin is based on the average GD value.

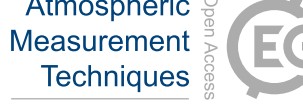



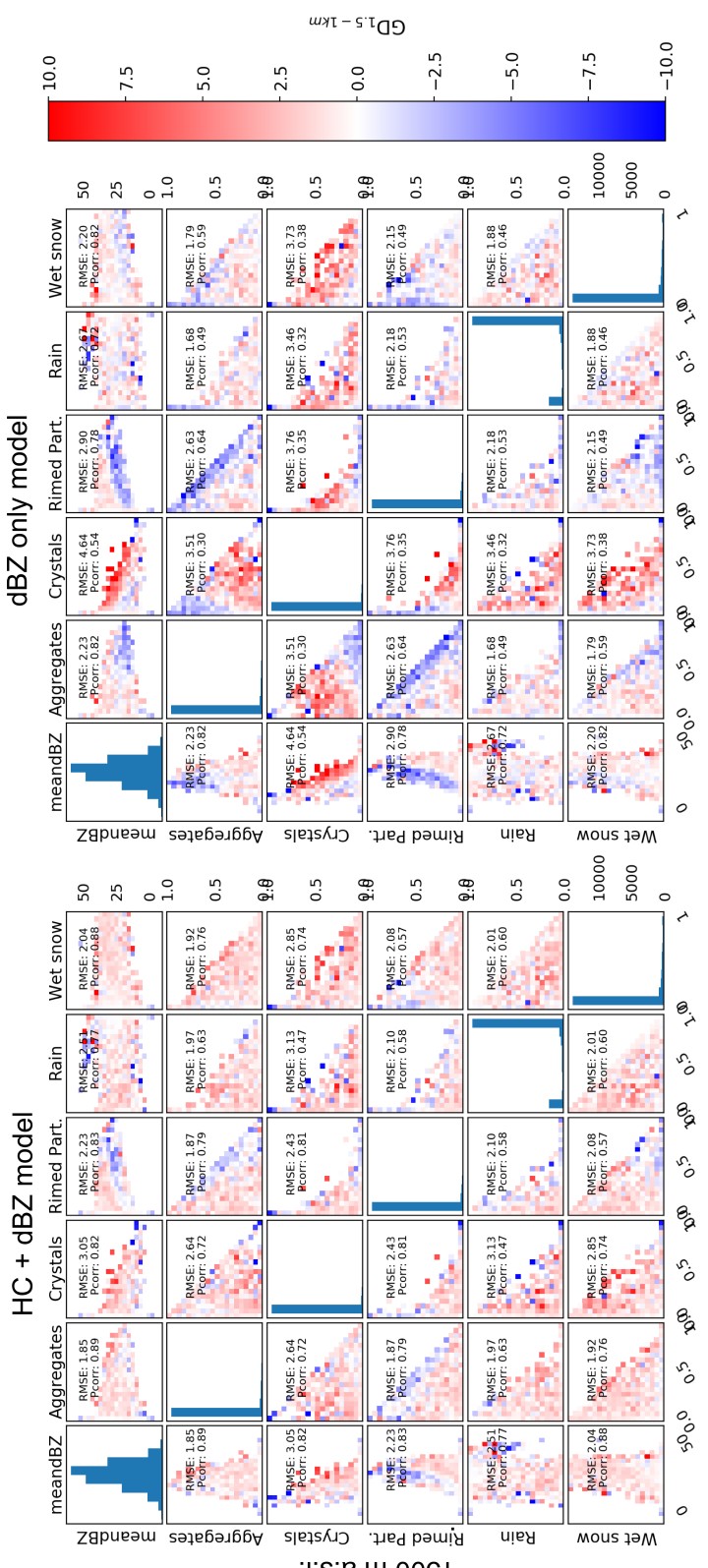

**Figure 8.** 2D error matrices from 1500 m a.s.l. for HC + dBZ-model (left) and dBZ-only model (right). The colour for each bin is based on the difference between the observed binned average and the binned average of the model output.



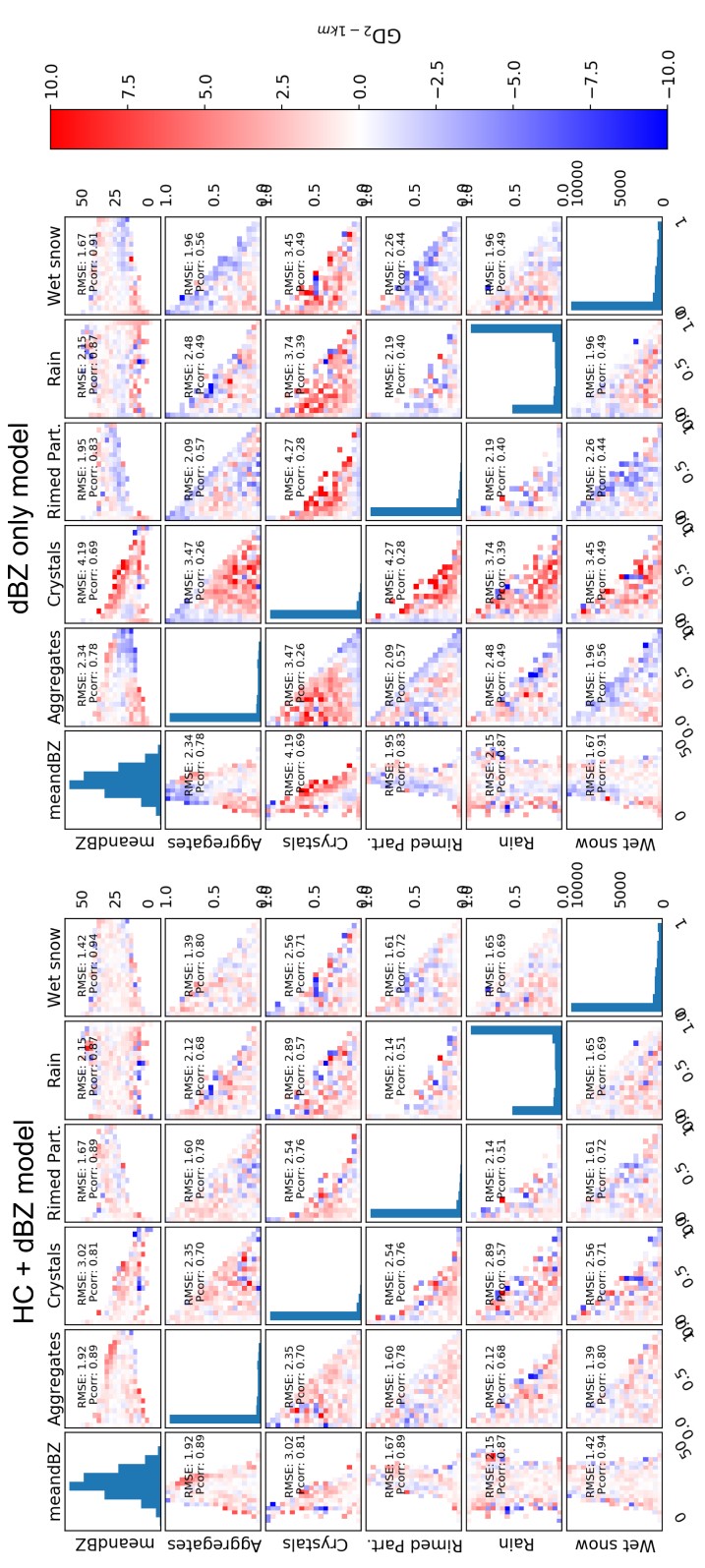

**Figure 9.** 2D error matrices from 2000 m a.s.l. for HC + dBZ-model (left) and dBZ-only model (right). The colour for each bin is based on the difference between the observed binned average and the binned average of the model output.





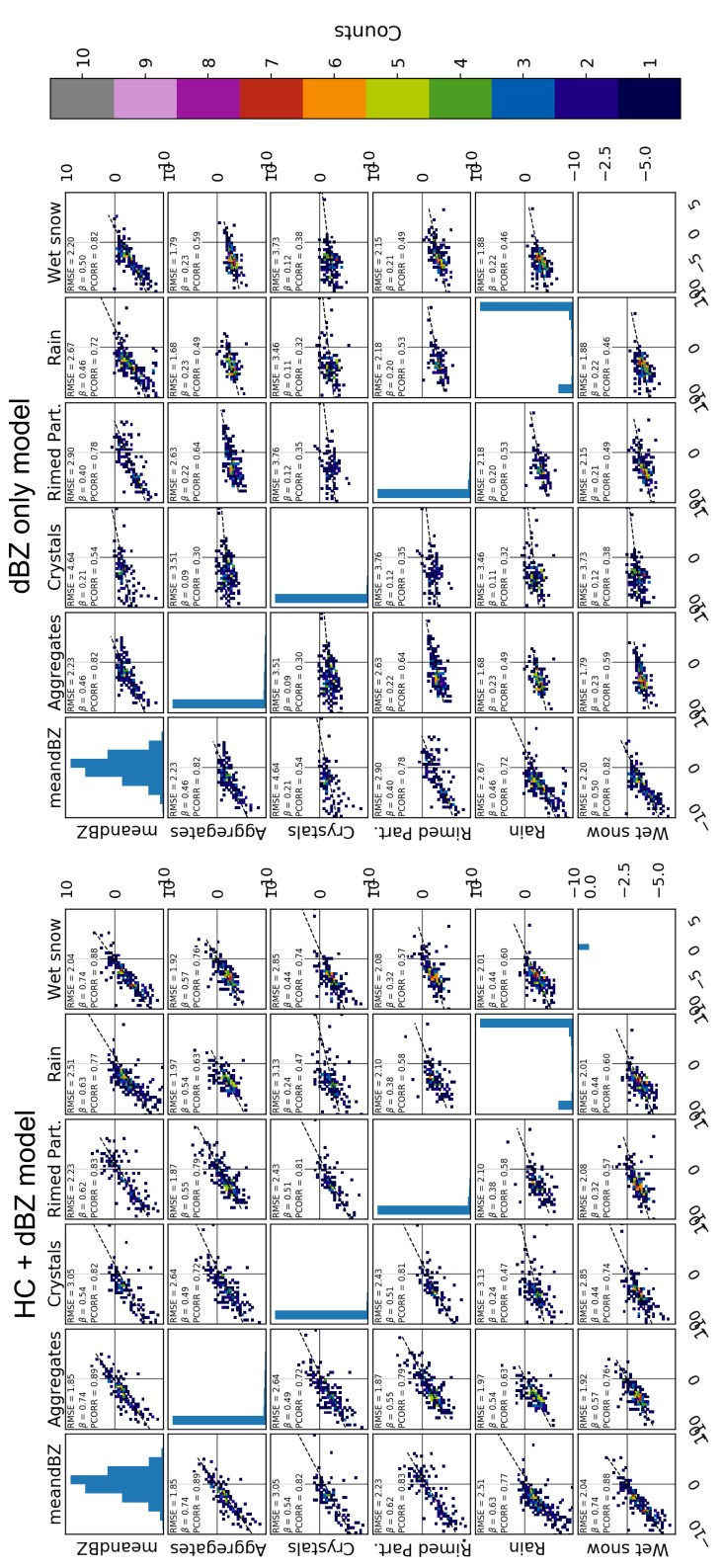

**Figure 10.** 2D matrices for observed (x-axes) and predicted (y-axes) GD for the HC + dBZ-model (left) and dBZ-only model (right) trained from 1500 m a.s.l. and data aloft. The number of points in each bin is indicated in colour.





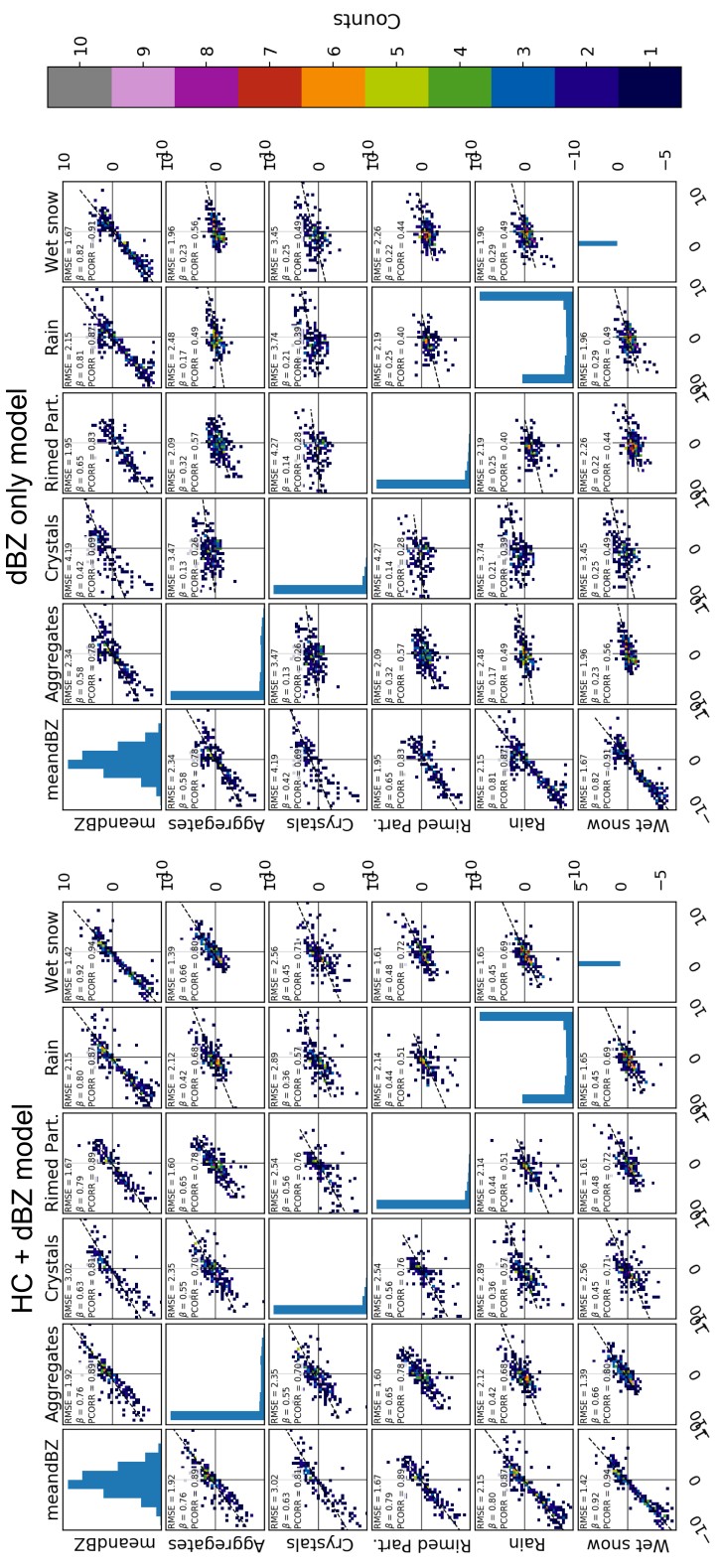

**Figure 11.** 2D histograms for observed (x-axes) and predicted (y-axes) GD for the HC + dBZ-model (left) and dBZ-only model (right) trained from 2000 m a.s.l. (bottom row) and data aloft. The number of points in each bin is indicated in colour.

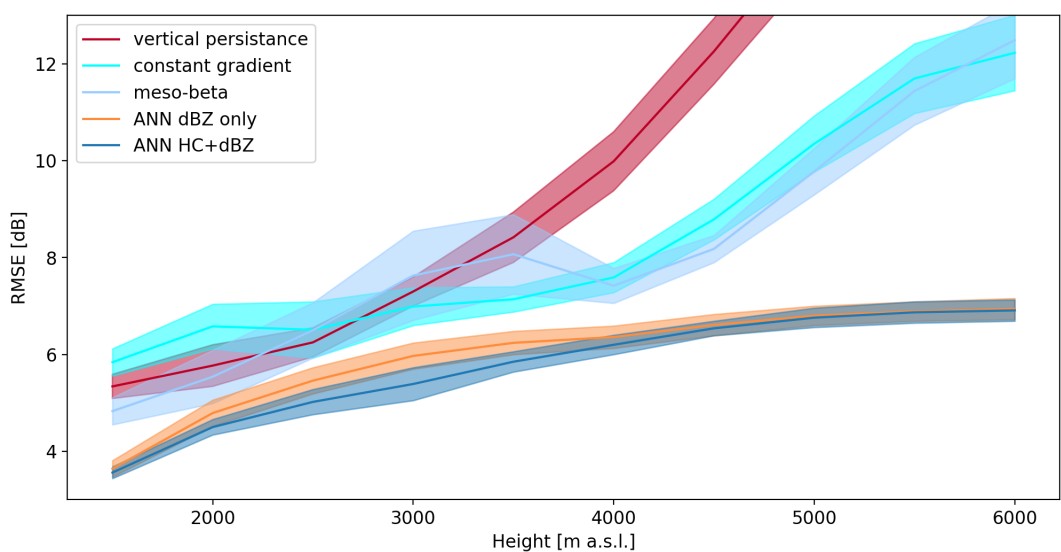

**Figure 12.** RMSE for traditional VPR models and for the ANN models trained with data starting from increasing altitude levels. The thick lines indicate the average values and the shading the quartiles calculated over ten model runs using different combinations of events for the training, validation and testing datasets.





**Figure 13.** Taylor diagrams of models trained from increasing altitudes. Each diagram shows the relative performance of the models making predictions from increasing altitude levels and in terms of the centered RMS error (green contours), correlation coefficient (azimuth angle) and standard deviation (radial distance).





**List of Tables**




**Table 1.** Median number of pixels for 500 m height intervals (boundaries in first column) at increasing distances from the radar based on geometry for a single volume scan and a cone with 4 km radius at the base and a 10 km radius at the top. The altitudes given in the first two columns are in metres above sea level.

| $h_b$ [m] | h [m] | 20-30 km | 40-50 km | > 60 km |
|---|---|---|---|---|
| 750 | | | | |
| 1250 | 1000 | 154 | 141 | 122 |
| 1750 | 1500 | 178 | 171 | 158 |
| 2250 | 2000 | 215 | 201 | 131 |
| 2750 | 2500 | 191 | 182 | 163 |
| 3250 | 3000 | 218 | 204 | 205 |
| 3750 | 3500 | 214 | 181 | 47 |
| 4250 | 4000 | 279 | 177 | 152 |
| 4750 | 4500 | 251 | 174 | 100 |
| 5250 | 5000 | 365 | 251 | 189 |
| 5750 | 5500 | 301 | 194 | 94 |
| 6250 | 6000 | 371 | 343 | 294 |
| 6750 | 6500 | 339 | 273 | 129 |
| 7250 | 7000 | 392 | 353 | 255 |
| 7750 | 7500 | 379 | 354 | 239 |
| 8250 | 8000 | 393 | 379 | 276 |
| 8750 | 8500 | 409 | 363 | 296 |
| 9250 | 9000 | 401 | 379 | 293 |
| 9750 | 9500 | 384 | 380 | 344 |
| 10250 | 10000 | 406 | 383 | 350 |





**Table 2.** Statistics for the precipitation events used in this study. Daily precipitation accumulations and wind speeds are from the ground station Cham at approximately 15 km distance from the Albis radar. The GWT weather type classification is also shown (more details in Weusthoff (2011)). LP, HP and FP represent High Pressure, Low Pressure and Flat Pressure situations respectively, the other acronyms are abbreviations of GWTWS flow directions. The letters A and C indicate advective or convective types.

| Date | GWTWS Type | Duration [hours] | max / $\mu$ Windspeed [m/s] | Daily Precip [mm] |
|---|---|---|---|---|
| 08-01-2016 | W (A) | 11 | 5.8 / 1.6 | 11.1 |
| 31-01-2016 | NW (A) | 10 | 11.4 / 2.7 | 11.1 |
| 23-02-2016 | W (A) | 7 | 12.1 / 3.2 | 7.8 |
| 02-03-2016 | W (A) | 7 | 21.0 / 2.35 | 7.1 |
| 17-04-2016 | SW (A) | 17 | 9.8 / 1.9 | 17.6 |
| 12-05-2016 | LP (C) | 21 | 10.4 / 2.6 | 31.9 |
| 23-05-2016 | SE (A) | 22 | 8.0 / 2.6 | 25.0 |
| 16-06-2016 | SW (A) | 12 | 7.5 / 1.9 | 20.9 |
| 12-07-2016 | SW (A) | 19 | 6.6 / 1.4 | 47.9 |
| 25-10-2016 | W (A) | 20 | 5.9 / 1.2 | 20.8 |
| 31-01-2017 | NW (A) | 23 | 5.7 / 1.3 | 34.7 |
| 09-03-2017 | NW (A) | 17 | 5.7 / 1.6 | 16.3 |
| 18-03-2017 | NW (A) | 11 | 9.8 / 1.7 | 11.1 |
| 25-04-2017 | W (A) | 15 | 10.7 / 1.9 | 24.4 |
| 12-05-2017 | LP (C) | 4 | 10.8 / 1.6 | 6.6 |
| 28-06-2017 | SW (A) | 14 | 11.0 / 2.1 | 26.3 |
| 10-07-2017 | SW (A) | 6 | 19.1 / 1.8 | 23.7 |
| 18-08-2017 | SW (A) | 4 | 9.9 / 1.5 | 35.7 |
| 31-08-2017 | SW (A) | 20 | 5.8 / 1.5 | 29.4 |
| 01-09-2017 | SW (A) | 16 | 7.5 / 2.6 | 24.2 |
| 12-11-2017 | W (A) | 9 | 23.5 / 3.3 | 14.6 |
| 22-01-2018 | NW (A) | 22 | 9.4 / 2.2 | 29.1 |
| 17-02-2018 | W (A) | 14 | 5.8 / 1.5 | 16.4 |
| 28-03-2018 | W (A) | 13 | 9.9 / 2.2 | 6.1 |
| 30-04-2018 | SW (A) | 3 | 15.3 / 2.4 | 1.2 |
| 14-05-2018 | E (A) | 7 | 6.8 / 1.7 | 9.6 |
| 15-05-2018 | N (A) | 9 | 7.0 / 2.0 | 7.6 |
| 22-05-2018 | FP (C) | 6 | 9.7 / 1.5 | 39.8 |
| 30-05-2018 | HP (C) | 2 | 9.5 / 1.8 | 0.0 |
| 04-06-2018 | FP (C) | 2 | 13.9 / 1.8 | 0.8 |