# Peer review of "Learning about the vertical structure of radar reflectivity using hydrometeor classes and neural networks in the Swiss Alps"

_Atmospheric Measurement Techniques, 2019_

## Referee Comment (RC1) · Anonymous Referee #1 · 22 Nov 2019

This paper aims to investigate a new localised vertical profile of reflectivity (VPR) correction technique by using machine learning. Radar reflectivity and hydrometeor proportions at equally spaced altitude from 30 precipitation events are taken as input for the training of an artificial neural network which then predicts the vertical change in reflectivity. This innovative method is able to provide a spatial variability for the VPR, tackling one of the major limitations of the standard techniques used operationally. The method outperforms the swiss operational VPR correction algorithm in predicting the growth and decay of precipitation during their fall.

The paper covers an important topic, has significant novelty and is well written. Consequently, I recommend its publication with some minor issues.

- General comments:

1) The shape of the VPR is particularly driven by the freezing level height. This latter also helps discriminate liquid and solid hydrometeors within the classification algorithm used (Besic et al., 2016). Can the worst performances of the "dBZ-only model" compared to the "HC + dBZ-model" be just attributed to the lack of information about the isotherm 0°C? In your opinion, would a "freezing level + dBZ-model" perform as well as the "HC + dBZ-model" or do the hydrometeor proportions bring valuable information?

2) Does the new method perform similarly in the different type of precipitation events studied (stratiform, convective)?

3) Please explicit in the text all variables used in the equations.

- Specific comments:

1) Section 2.1: Are the radar data time synchronized?

2) Figure 4: Median reflectivities are weaker in the lower atmosphere (<4000 m) in the convective event than in both other cases displayed. How do you explain this ? Is it due to the presence of large stratiform precipitation areas?

3) Section 4.2: Do you have any explanation of why the dBZ-only models have difficulty with predicting growth values higher than 10 dB?

- Technical corrections:

1) p4 l22-24: For more clarity, I suggest to write "-10 to 60 dBZ" for the reflectivity range as well as the following radar variable ranges.

2) p4 l26: "ice, hail or high density graupel (IH/HDG)"

3) p5 l7: "a range of unidirectional wind speeds..."

4) p5 l8: "3 km height"

5) Figures 7, 8, 9, 10 and 11: labels of lower x-axis as well as right y-axis overlap

---

## Referee Comment (RC2) · Hidde Leijnse (Referee) · 14 Jan 2020

This paper describes a novel method for estimating the local vertical growth and decay of precipitation based on an artificial neural network (ANN). Two ANNs are investigated: one based only on dBZ profiles, and one based on profiles of both dBZ and the result of a hydrometeor classification scheme. It is shown that the method outperforms other vertical profile correction methods commonly used for operational weather radar data. I think the paper is well-written and the topic is highly relevant because the vertical structure of precipitation remains an issue with operational weather radar data. The paper is novel in two aspects: 1) the use of cones to extract horizontally distributed

vertical precipitation variation information, and 2) the use of ANNs to correct for this variability. I did not find many things that I wanted to have clarified in the paper, and hence I think the paper can be published after (very) minor revisions. Some specific comments are provided below.

**Specific comments**

- You use polarimetric variables through a hydrometeor classification scheme. You could also use the polarimetric variables directly in the ANN. What is the reason for going thought the HC scheme in this study? I think that a short discussion on this could be added to the paper.

- How would this method be implemented operationally? It would really help me to have some sort of short explanation (possibly including a graphical representation of the implementation) on how one would derive 2-dimensional precipitation information on the ground from 3-dimensional volume scans from a radar using this method.

- If there are two or more radars that both cover the same area, could information from all of these be used to improve results? Consider discussing this briefly in the concluding section.

- On p.5, you state that the 30-minute time scale is needed in order for the upper and lower part of the vertical precipitation profiles to be linked. I can follow this reasoning, but it would be good to provide some more quantitative arguments to support this (like using the fall velocity of precipitation particles).

- On p.10, line 32, you introduce the regression slope $\beta$. Can you briefly introduce how this is defined? I find it puzzling that you say that there are high regression slopes, and then say that $\beta < 0.54$. Hence my question of defining $\beta$.

---

## Author Comment (AC1) · 30 Mar 2020

We kindly thank Hidde Leijnse and the anonymous referee for their valuable reviews and their highly relevant questions and comments for the improvement of the manuscript. Our answers and suggested adaptations are specified below.

Anonymous referee

This paper aims to investigate a new localised vertical profile of reflectivity (VPR) correction technique by using machine learning. Radar reflectivity and hydrometeor proportions at equally spaced altitude from 30 precipitation events are taken as input for

the training of an artificial neural network which then predicts the vertical change in reflectivity. This innovative method is able to provide a spatial variability for the VPR, tackling one of the major limitations of the standard techniques used operationally. The method outperforms the swiss operational VPR correction algorithm in predicting the growth and decay of precipitation during their fall. The paper covers an important topic, has significant novelty and is well written. Consequently, I recommend its publication with some minor issues.

General comments:

1) The shape of the VPR is particularly driven by the freezing level height. This latter also helps discriminate liquid and solid hydrometeors within the classification algorithm used (Besic et al., 2016). Can the worst performances of the "dBZ-only model" compared to the "HC + dBZ-model" be just attributed to the lack of information about the isotherm $0°C$? In your opinion, would a "freezing level + dBZ-model" perform as well as the "HC + dBZ-model" or do the hydrometeor proportions bring valuable information?

This is a very interesting point and one might indeed wonder whether the isotherm $0°C$ information is dominant in the results. The reason why an explicit description of the isotherm $0°C$ was not used as an ANN input is because no melting layer detection algorithm has yet been successfully applied to the operational C-band radar data. We also wanted to use as much as possible the radar data instead of data from external sources. As a compromise, information on the isotherm $0°C$ is included through the hydrometeor classification algorithm, which uses a measure of the vertical distance from the isotherm $0°C$, as well as polarimetric radar variables, to distinguish a liquid/melting/ice phase indicator. In practice, this means that the wet snow hydrometeors often indicate the presence of a melting layer (which is not the same as the isotherm $0°C$, but more valuable for VPR correction purposes). To consider your comment, we performed additional experiments using reflectivity+wet snow proportions ANN models to verify the influence of this melting layer "proxy" on the results. The improvements compared to a dBZ-only model were very modest (see Figure 1 below).

[Figure]

2) Does the new method perform similarly in the different type of precipitation events studied (stratiform, convective)?

This is a very interesting question. Some efforts were made earlier in the study to stratify the training of the ANN models, i.e. to train one ANN for convective, and a different one for stratiform events. Events were classified based on the GrossWetterTypes (GWTWS) weather types, based on convective available potential energy (CAPE) and the characteristic time scale with which CAPE is removed at an exponential rate by convection (TAU), as well as based on visual analysis. However, none of these classifications turned out conclusive, and indeed, many of the convective cases actually consisted of convective cells which were embedded in stratiform precipitation events. The purely convective "cones" were therefore too sparse to train a separate ANN or to confidently calculate error statistics. The effect of the possible over representation of stratiform rainfall events in the dataset is also mentioned in lines 12-14 on page 12.

3) Please explicit in the text all variables used in the equations.

This has been added at all the relevant locations.

- Specific comments:

1) Section 2.1: Are the radar data time synchronized?

Does the referee refer to synchronization in absolute way, between elevations or between radars in the network? Each single sweep of each radar is synchronized (meaning that there is a time stamp at the beginning and at the end of each sweep). Naturally, it is impossible to measure at different elevations at exactly the same time. We are aware that in some studies, efforts have been made to "synchronize" between elevations by advecting measurements forward or backward with optical flow. However, this is not straightforward to do in polar coordinates and was not applied in the context of this study.

We suggest the following addition: page 4, line 7; "Swiss operational network (all time

synchronised)"

2) Figure 4: Median reflectivities are weaker in the lower atmosphere (<4000 m) in the convective event than in both other cases displayed. How do you explain this ? Is it due to the presence of large stratiform precipitation areas?

Yes, large stratiform precipitation areas and/or some attenuation may partly explain this. Another explanation is related to the intermittent and fast-moving nature of convective systems. The cones were not necessarily sampled in the core of the convective system, and as such these may be partially filled with much lower values or no values. Even though cones which were less than 10% filled in their bottom 4000 meters were removed from the data set and the median value is a more robust statistic than the average, it may still be influenced by a smaller number of samples. Also, these typical profile examples were taken over the entire spatial domain and for a single 30 minute time step. Another time step from the same event, shows higher values in the lower atmosphere (Figure 2 below). This image was not selected because in this case the reflectivity profile could be more easily confused with a stratiform profile.

3) Section 4.2: Do you have any explanation of why the dBZ-only models have difficulty with predicting growth values higher than 10 dB?

The observed growth and decay values are normally distributed between -10 and 10 dB with tails which extend to -20 and 20 dB. Figure 7 shows at 2000 m a.s.l. (similar behaviour at 2500 m a.s.l.) that these "pockets" of higher growth values are (among other things) related to high proportions of aggregates (Figure 7, second row, first column matrix and within this matrix the first row, second column – see also indications in Figure 3 below). It can be seen in the matrix in the middle column that the HC + dBZ model reproduces these pockets, while the dBZ-model does not (last column). Possibly, the dBZ-only ANN considers these values as outliers and smooths the results approximately within -10 to 10 dB values whereas the HC + dBZ ANN can actually identify the relationship with the high proportions of aggregates and models these.

See also page 10, line 14-17: "However, more specific patterns can also be observed, such as the pronounced growth values for cones with low to moderate reflectivity values at altitude h and high proportions of aggregates or any presence of rimed particles at that same altitude h. While these patterns are also visible in the HC + dBZ-model output, the dBZ-only model is unable to reproduce them."

Suggested addition to this line (page 10, lines 17-19): "This is probably because the dBZ-only model does not have the necessary predictors to explain this variability and thus treats the high GD values as outliers, while the HC + dBZ model, with the additional information on hydrometeor proportions, can recognise these patterns."

- Technical corrections:

1) p4 l22-24: For more clarity, I suggest to write "-10 to 60 dBZ" for the reflectivity range as well as the following radar variable ranges.

Now changed from "-" to "to".

2) p4 l26: "ice, hail or high density graupel (IH/HDG)"

Now changed.

3) p5 l7: "a range of unidirectional wind speeds..."

Now changed.

4) p5 l8: "3 km height"

Now changed.

5) Figures 7, 8, 9, 10 and 11: labels of lower x-axis as well as right y-axis overlap

Now changed.

Hidde Leijnse (Referee)

This paper describes a novel method for estimating the local vertical growth and decay

of precipitation based on an artificial neural network (ANN). Two ANNs are investigated: one based only on dBZ profiles, and one based on profiles of both dBZ and the result of a hydrometeor classification scheme. It is shown that the method outperforms other vertical profile correction methods commonly used for operational weather radar data. I think the paper is well-written and the topic is highly relevant because the vertical structure of precipitation remains an issue with operational weather radar data. The paper is novel in two aspects: 1) the use of cones to extract horizontally distributed vertical precipitation variation information, and 2) the use of ANNs to correct for this variability. I did not find many things that I wanted to have clarified in the paper, and hence I think the paper can be published after (very) minor revisions. Some specific comments are provided below.

Specific comments

1) You use polarimetric variables through a hydrometeor classification scheme. You could also use the polarimetric variables directly in the ANN. What is the reason for going thought the HC scheme in this study? I think that a short discussion on this could be added to the paper.

This is a very good question. The motivation is that the information provided by the hydrometeor classification is somehow already "filtered" from the noise in the estimated radar variables because it contains additional information coming from the use of scattering simulations to identify which clusters correspond to which hydrometeor types. From a physical point of view, one may also expect that hydrometeor class information can better describe the processes involved in the growth and decay of precipitation than polarimetric variables as such.

We suggest the following addition (page 4, lines 29-34): "The motivation to use hydrometeor proportions as input for the ANN models rather than polarimetric variables is twofold. On the one hand, the hydrometeor classification is already filtered from the noise in the estimated radar variables because it contains additional "physical" information coming from scattering simulations to identify which clusters correspond to which hydrometeor types. On the other hand, from a physical point of view, hydrometeor class information can better describe the processes involved in the growth and decay of precipitation than the raw polarimetric variables."

2) How would this method be implemented operationally? It would really help me to have some sort of short explanation (possibly including a graphical representation of the implementation) on how one would derive 2-dimensional precipitation information on the ground from 3-dimensional volume scans from a radar using this method.

Thank you for this question. Indeed, the proposed vertical profile correction technique is part of a larger QPE scheme and it is valuable to discuss the possibilities for operational implementation of the method.

We suggest the following addition: (page 15, lines 3-21): "The requirements for the operational implementation of a new vertical profile correction technique are stringent and so the potential of the proposed method should also be evaluated in the light of these requirements. Firstly, an operational correction method should be able to function at all times. The method proposed in this study could fail if for some reason one of the polarimetric variables is unavailable or compromised. In such cases, the dBZ-only model could substitute the HC + dBZ model. However, swapping models may also lead to discontinuities from one radar image to the next, and some temporal aggregation may be necessary to resolve such issues. In terms of processing costs, once the cones are extracted and the model is trained, the application of the ANN models to existing data should be relatively fast. The ANNs could be re-trained and tested regularly after hardware changes to the radar system and with newly available high-quality data. It may also be considered to train the model and apply the correction to larger scales in some regions such as the Swiss plateau and to smaller scales in other regions such as the Alps. Within each area, the appropriate ANN model (1500 m asl, 2000 m asl) can be applied to the lowest or best available radar elevation (and the data from aloft) in order to estimate the GD towards the ground level. Because the
method is based on hydrometeor classification data rather than polarimetric variables, the output of the classification scheme is more consistent between different radars, and the ANN model can be applied in regions where more than one radar cover the same area. Finally, the GD term is added to the lowest available reflectivity measurement to estimate reflectivity values, and ultimately precipitation rates at the ground. Operational implementation of this technique still requires further study and improvements. Nevertheless, the approach proposed in this study takes advantage of the capability of ML techniques to learn complex non-linear relationships between polarimetric radar variables (represented by the HC proportions) along the vertical column. It demonstrates their potential to improve the extrapolation of high altitude radar observations to lower levels, which is a relevant step for the improvement of polarimetric radar QPE in complex terrain."

3) If there are two or more radars that both cover the same area, could information from all of these be used to improve results? Consider discussing this briefly in the concluding section.

For this study, the ANN models have been trained with data within the well-visible region of the Albis radar. The overlapping with other radars occurs at distances further away from the Albis radar than the considered region. Because the training of the ANN requires a ground "truth" for verification, it would be complicated to use data at further distances from the radar for this.

However, once the ANN models are applied to observations in order to extrapolate to the ground level, using the lowest/best available data from multiple radars could definitely improve the results. This could be done in different ways; one option would be to merge the predictors before passing them to the ANN. The additional value of using hydrometeor classification data rather than polarimetric variables in this case, is that the output of the classification scheme is more consistent between radars. This would still require a sound strategy to deal with the merging of the hydrometeor proportions. Another option would be to apply the ANN to the different radars and then average

the GD values, e.g. based on the respective visibilities if the radars. See the addition proposed in the question above.

4) On p.5, you state that the 30-minute time scale is needed in order for the upper and lower part of the vertical precipitation profiles to be linked. I can follow this reasoning, but it would be good to provide some more quantitative arguments to support this (like using the fall velocity of precipitation particles).

Suggested addition (page 5 lines 26–29): "The 10 km and 30 minute scales have been selected because it is expected that at these scales the lower part of the VPP can be related to the VPP and hydrometeor proportions aloft: if precipitation falls at an average speed of 5 m s-1 (slower above and faster below the melting layer), it would cover a vertical distance of 10 km in about 30 minutes."

5) On p.10, line 32, you introduce the regression slope $\beta$. Can you briefly introduce how this is defined? I find it puzzling that you say that there are high regression slopes, and then say that $\beta$ <0.54. Hence my question of defining $\beta$

In this case, $\beta$ measures the degree of conditional bias with respect to the observations. It is given by the formula:

$\beta$ = ($\sigma$pred / $\sigma$obs) *

Where  is the correlation coefficient between predictions and observations, and $\sigma$pred and $\sigma$obs are the corresponding standard deviations. In modelling studies, the regression slope is typically calculated with respect to the predictions (predictions on x axis), resulting in a $\beta$ > 1 if the standard deviation of the predictions is smaller than the standard deviation of the observations. Because machine learning methods rely on error minimization principles there is often a loss of variance in the predictions with respect to the observations. As such, $\beta$ is typically lower than 1 rather than larger than 1.

However, the sentence is still ill-formulated. We suggest the following adaptations :

(page 11, lines 6-15): "Figures 10 and 11 are the same as Fig. 5, but, instead of

comparing the instantaneous predicted and observed growth and decay values, it compares the binned averages of Fig. 7, and Figs. 8 and 9. The RMSE, Pearson correlation coefficient and the regression slope $\beta$ are given. In this study, $\beta$ measures the degree of conditional bias with respect to the observations. It is given by the formula:

$\beta = (\sigma\text{pred} / \sigma\text{obs})$ *

Where  is the correlation coefficient between predictions and observations, and $\sigma$pred and $\sigma$obs are the corresponding standard deviations. In modelling studies, the regression slope is typically calculated with respect to the predictions, resulting in a $\beta > 1$ if the standard deviation of the predictions is smaller than the standard deviation of the observations. Because machine learning methods rely on error minimization principles there is often a loss of variance in the predictions with respect to the observations. As such, the $\beta$ calculated in this study with respect to the observations is typically lower than 1.

As expected, the correspondence of the average values is much better than the instantaneous ones.."

And:

(page 11, lines 17-20): "For the HC + dBZ-model the observed and predicted GD for combinations of average reflectivity and any hydrometeor class show good agreement with relatively low regression slopes ($0.54 < \beta < 0.74$) and high correlation coefficients ($> 0.77$). The worst performances, for the crystals and rain classes at 1500 m asl, are clearly related to a few outliers (Fig. 10).

———————————————

[Figure]

[Figure]

**Fig. 1.**

[Figure]

**Fig. 2.**

[Figure]

Observations ZH + Hydrometeors ZH only

1500 m asl

2000 m asl

**Fig. 3.**